# Defense Against Multi-target Backdoor Attacks

## Abstract

Neural Trojan/Backdoor attacks pose a significant threat to the current deep-learning-based systems and are hard to defend against due to the lack of knowledge about triggers. In this paper, we first introduce a variant of BadNet that uses multiple triggers to control multiple target classes and allows these triggers to be at any location in the input image. These features make our attack more potent and easier to be conducted in real-world scenarios. We empirically found that many well-known Trojan defenses fail to detect and mitigate our proposed attack. To defend against this attack, we then introduce an image-specific trigger reverse-engineering mechanism that uses multiple images to recover a variety of potential triggers. We then propose a detection mechanism by measuring the transferability of such recovered triggers. A Trojan trigger will have very high transferability i.e. they make other images also go to the same class. We study many practical advantages of our attack and then apply our proposed defense mechanism to a variety of image datasets. The experimental results show the superiority of our method over the state-of-the-arts.

## 1 Introduction

Deep learning models have been shown to be vulnerable to various kinds of attacks such as adversarial attacks proposed by Goodfellow et al. (2015); Moosavi-Dezfooli et al. (2017); Fawzi et al. (2018), and backdoor or Trojan attacks as discussed in Ji et al. (2017); Gu et al. (2017); Chen et al. (2017); Chan & Ong (2019). Among them, Trojan attacks Gu et al. (2017); Liu et al. (2017) are one of the hardest to defend against. In its simplest form, in Trojan attacks, the data for training a model is poisoned by putting a small trigger in it Gu et al. (2017). A model trained on such a poisoned dataset behaves expectedly with pure data but would wrongly predict when a test data is poisoned with the trigger. A small trigger may not cause any issue with other non-Trojan models or human users, and thus may escape detection until it is able to cause the intended harm. Due to its stealth, detection of Trojan backdoors requires specialised testing.

Testing methods to detect Trojan attacks rely heavily on the assumed threat model. For Badnet Gu et al. (2017; 2019), with only one target class with a fixed trigger location, Neural Cleanse (NC) Wang et al. (2019), and GangSweep (GS) Zhu et al. (2020) provide useful defense through trigger reverse engineering. NC requires an optimization process over a large set of pure images for the trigger reverse engineering. Moreover, both NC and GS rely on detecting a single Trojan trigger as anomalous patterns. Unsurprisingly, they do not work in multi-target attack settings as Trojan triggers are many and thus are not anomalous. STS Harikumar et al. (2021) can defend against triggers placed anywhere, and does not involve anomaly detection, but fails if the number of images used during trigger reverse-engineering is small, as mentioned by the authors. STRIP Gao et al. (2019), Grad-CAM Selvaraju et al. (2017), Neural Attention Distillation (NAD) Li et al. (2021), and Fine-pruning (Fp) Liu et al. (2018); Zhao et al. (2020) are some of the remedial measures that try to de-risk the use of a potential Trojan model either by trying to create a pure model out of it (NAD, Fp) or to stop Trojan images during test time (STRIP, Grad-CAM). NAD and Fine-pruning fail if the features of the trigger overlap with that of the pure images or if we do not have a sufficient number of images to separate the trigger features from other features. STRIP fails if the location of the trigger overlaps with the main features of the pure images. *Hence, a proper testing method for multi-target Trojan attack with minimal restriction on triggers and that can work in sample-poor settings is still an open problem.* We focus on fixed trigger-based Trojans because it is much more robust and feasible than the more recent

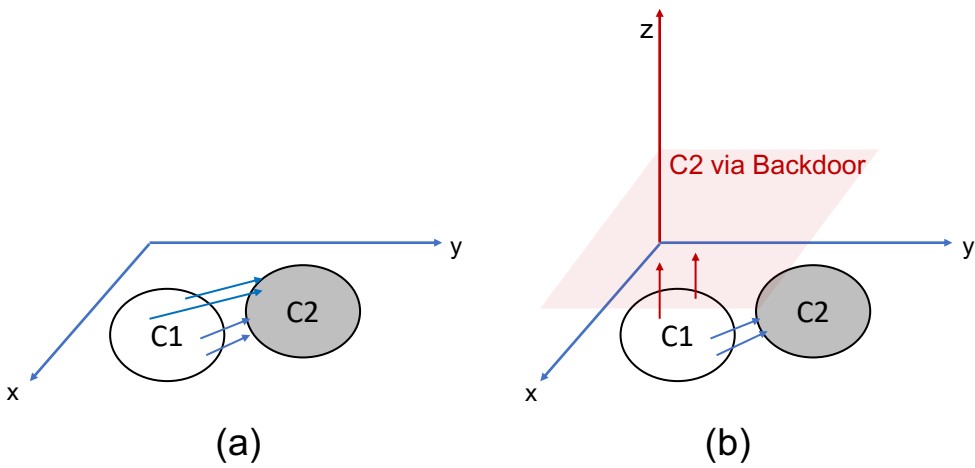

Figure 1: (a) Pure model when backdoor is not present. Each image from class C1 has different perturbations (blue arrows) to be classified as class C2. (b) When the backdoor is present it creates a shortcut subspace (red plane). Some of the images from C1 will find perturbations that are now aligned with this backdoor subspace (red arrows) and thus are Trojans. All the Trojan triggers (single-target setting) are similar, and they can take any image from C1 to the C2 via the backdoor. Some images of C1 will still find image-specific perturbations (blue arrows) because they are closer to the original subspace of C2 (gray) than the backdoor subspace.

input-aware attacks (IA) Nguyen & Tran (2020a;b). IA generates perturbations for each individual image and in applications like autonomous car where a sensor makes multiple measurements of the same object at slightly different poses, the input-aware attacks would likely fail to influence the composite decision process, and thus, in our opinion, they do not pose a significant threat.

Our objective is to create a defense for multi-target Trojan attacks, with minimal assumptions about the trigger, e.g., the trigger can be placed anywhere, and we may not have lots of pure images during the Trojan model detection process. Our proposed method is built on trigger reverse engineering but designed in a way to detect multi-target Trojan attacks. The intuition of our method is illustrated in Figure 1 through an understanding of the classification surface in pure and Trojan models. In pure models, each image from class C1 has different perturbations (blue arrows) to be classified as class C2 (Figure 1a). But in Trojan models, due to the presence of the trigger, it creates a shortcut subspace (backdoor), shown as a red shaded plane (e.g., $z = 1$) in Figure 1b. Some of the images from C1 will find perturbations that are now aligned with this backdoor subspace (red arrows) and thus are Trojan triggers. These Trojan triggers are similar and thus transferable because they will make any image from C1 go to C2 through the backdoor subspace. For some images, however, these backdoor perturbations are larger than directly going from C1 to C2, and thus their perturbation will remain image-specific (blue arrows in Figure 1b). Our method is based on finding these transferable perturbations in a given model because their existence indicates the presence of a backdoor. We do this in two steps: perform trigger reverse engineering on each image in the set of pure images (*Data_trigger*) and then verify their transferability based on a separate set of pure images (*Data_transfer*). To perform trigger reverse engineering, we solve an optimization problem to find a small perturbation that takes an image to each of the other classes. Thus, for a 10-class problem, each image will generate 9 triggers. Each reverse-engineered trigger is then pasted on the images from *Data_Transfer*, and the entropy of the class distribution is measured for each trigger. A Trojan trigger would cause most of the images to go to the same class, thus resulting in a skewed class distribution, which means a small entropy. We provide a mechanism to compute a threshold for the entropy below which a perturbation can be termed a Trojan trigger. Our method can work with small size *Data_trigger* set as we would assume that for small trigger trained Trojan models most of the images will take the shortcut subspace for crossing the class boundary. We call our proposed attack as **M**ulti-**T**arget **T**rojan **A**ttack (**MTTA**) and the associated defense as **M**ulti-**T**arget **D**efense (**MTD**.

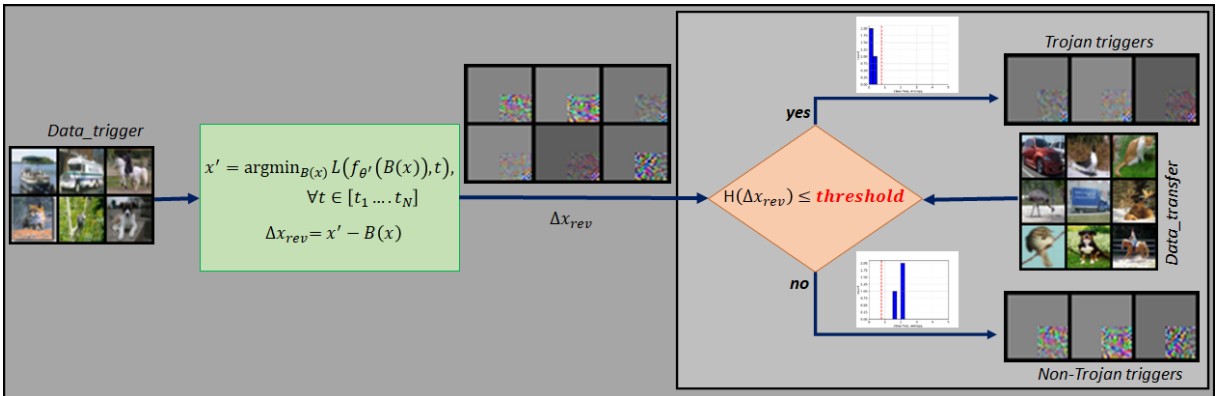

Figure 2: Schematic diagram of the proposed Multi-target Defense (MTD) method. Images from the Data_trigger is used for trigger reverse engineering. The reverse-engineered triggers are tested on Data_transfer to check their transferability. Triggers that produces low entropy for the class distribution are termed Trojan triggers (top row) and the others are non-Trojan triggers (bottom row). The dotted red line in the entropy plots separate the Trojan and non-Trojan triggers. The original trigger used is a checkerboard pattern, thus the Trojan triggers contains a similar pattern (please zoom in to see the pattern).

To some extent our trigger reverse engineering process is similar to GangSweep, but instead of learning a GAN Goodfellow et al. (2014) we use individual triggers in the detection process. Because we check for Trojan in each class individually, our method works well even when all the classes are Trojaned. We show the efficacy of our method on four image datasets (MNIST, CIFAR-10, GTSRB, and Youtube Face). Additionally, we also show that our proposed attack is more robust than the well-known Badnet and input-aware attacks.

## 2  Method

A Deep Neural Network (DNN) can be defined as a parameterized function $f_\theta : \mathcal{X} \to \mathbb{R}^C$ that recognizes the class label probability of an image $x \in \mathcal{X}$ where $\mathcal{X} \sim P_\mathcal{X}$ and $\theta$ represents the learned function's parameter. Let us consider the probability vector for the image $x$ by the function $f_\theta$ as $[p_1....p_C]$, thus the class corresponding to $x$ will be $\operatorname{argmax}_{i \in [1..C]} p_i$.

### 2.1  Threat Model

The attack setting has three key features: 1) multiple triggers, $[\Delta x_1....\Delta x_N]$; 2) multiple target classes, $[t_1....t_N]$; and 3) trigger can be placed anywhere in the image. We use a square patch as trigger which when put on the image cause misclassification. However, the attacker can use triggers of any shape as long as it is not covering a large part of the whole image. We have different triggers associated with each target class and each pixel of the trigger has a different color. The target classes are a subset of classes randomly chosen from the known set of classes of the dataset i.e. $N < C$. The practicality of this trigger anywhere lies in the fact that in the real world an attacker can put a sticker on any location of the image, instead of carefully positioning it like Badnet. Mathematically, for the Trojan model the original DNN model with model parameters $\theta$ will be replaced by Trojan model parameters $\theta'$ denoted as $f_{\theta'}(.)$. The following shows the composition method for the trigger and the images.

***Definition 1*** : A trigger is formally defined as a small square image patch $\Delta x$ of size $s$ that is either physically or digitally overlaid onto the input image $x$ at a location $(i^*, j^*)$ to create a modified image $x'$. Concretely, an image of index $k$ of the dataset $x_k$ is altered into an image $x'_k$ by,

$$x'_k(i,j) = \begin{cases} a & \text{if } i \in [i^*, i^* + s], j \in [j^*, j^* + s] \\ b & \text{elsewhere.} \end{cases} \tag{1}$$

---

**Algorithm 1** Multi Target Defense (MTD).

---

**Inputs** : $x$, $C$, $f_{\theta'}(.)$, $x_{test}$, threshold
**Outputs**: target_classes, **Boolean** trojan_model
**for** each *class* in $C$ **do**
    Compute optimised image, $x'$ with *class* using Eq 3.
    Compute reverse engineered trigger, $\Delta x_{rev}$ with Eq 4.
    Compute entropy, $H(\Delta x_{rev})$ using $x_{test}$ with Eq 5.
    **if** $(H(\Delta x_{rev})) \leq$ threshold **then**
        target_classes.append(*class*)
    **end if**
**end for**
**if** length (target_classes) $\geq 1$ **then**
    trojan_model = **True**
**else**
    trojan_model = **False**
**end if**

---

where $a = (1 - \alpha(i', j'))x_k(i, j) + \alpha(i', j')\Delta x(i', j')$ , $b = x_k(i, j)$, $(i', j')$ denote the local location on the patch $(i', j') = (i - i^*, j - j^*)$ as defined in Harikumar et al. (2021). The transparency of the trigger is controlled by the weight, $\alpha$. This parameter can be considered as a part of the trigger, and we will be inclusively mentioning it as $\Delta x$. Meanwhile, the rest of the image is kept the same. In our setting, $(i^*, j^*)$ can be at any place as long as the trigger stays fully inside the image.

## 2.2 Trojan Detection

We use the validation dataset of pure images for trigger reverse engineering and transferable trigger detection by splitting it into two separate datasets: a) *Data_Trigger* - for trigger reverse engineering, and b) *Data_Transfer* - for checking transferability of the reverse engineered triggers. For each image in the *Data_Trigger* we find a set of perturbations by setting each class as a target class. Each reverse engineered trigger is then used on the images of *Data_Transfer* to compute the class-distribution entropies. If a perturbation is the Trojan trigger, then it will transfer to all the images and the class distribution would be peaky at the target class, resulting in a small entropy value. We provide a mechanism to compute the entropy threshold below which a perturbation is termed Trojan trigger.

### 2.2.1 Trigger Reverse Engineering

Given an image, $x \in \mathbb{R}^{Ch \times H \times W}$, where $Ch, H, W$ are the number of channels, height, and width with a target label $y$, we define $B(x)$ as the mask that only keeps inside pixels active for the optimization i.e.,

$$B(x) = x \odot B, \tag{2}$$

where $\odot$ is the element-wise product and B is a binary matrix. B has a value of 1 across a region $H/4 \times W/4$ across all $Ch$ channels, and can be positioned anywhere, as long as it is fully within the image. Alternatively, one can use a mask of size $H \times W$ with L1-regularization Park & Hastie (2007) on $B(x)$ (initial $x'$) to control the sparsity level of perturbations. We have shown some analysis based on both types of mask in the experiments (Section 3.7). We then minimize the cross-entropy loss between the predicted label for $B(x)$ and the target label $y$:

$$x' = \text{argmin}_{B(x)}\mathcal{L}\left(f_{\theta'}(B(x)), y\right). \tag{3}$$

We denote the new optimized value of $B(x)$ as $x'$. The reverse engineered trigger which we denote as $\Delta x_{rev}$ is the difference between $x'$ and $B(x)$:

$$\Delta x_{rev} = x' - B(x). \tag{4}$$

| Dataset | #classes | Input Size | Classifier | #Target classes | Pure accuracy | | | Trojan accuracy | |
|---------|----------|-----------|-----------|-----------------|-----------------|---|---|------------------|---|
| | | | | | Pure model | Trigger size | | Trigger size | |
| | | | | | | 4×4 | 8×8 | 4×4 | 8×8 |
| MNIST | 10 | 1×28×28 | 2 conv, 2 fc | 7 | 99.53 | 98.83 | 99.24 | 99.76 | 99.98 |
| GTSRB | 43 | 3×32×32 | PreActRes18 | 30 | 98.85 | 98.84 | 98.76 | 100.0 | 100.0 |
| CIFAR-10 | 10 | 3×32×32 | PreActRes18 | 7 | 94.55 | 93.93 | 94.39 | 100.0 | 100.0 |
| YTF | 1283 | 3×55×47 | Resnet18 | 384 | 99.70 | 99.55 | 99.34 | 96.73 | 99.79 |

Table 1: Dataset, classifier and MTTA attack configuration Pure accuracy of Pure models and MTTA Trojan Models as well as the Trojan accuracy of the MTTA Trojan models.

### 2.2.2 Transferability Detection

To check for transferability, we compute the entropy Shannon (1948) of the class distribution for each reverse engineered trigger when used on all the images of the *Data_transfer* as follows,

$$\mathrm{H}(\Delta x_{rev}) = -\sum_{i=1}^{C} p_i \log_2(p_i), \tag{5}$$

where $\{p_i\}$ is the class probability for the $i$-th class for using that perturbation. The entropy of a Trojan trigger will be zero if the Trojan attack success rate is 100%. However, in real-world situation, we assume a slightly less success rate that leads to a non-zero entropy value. The following lemma shows how to compute an upper bound on the value of this score for the Trojan models in specific settings, which then can be used as a threshold for detecting Trojans.

***Lemma 1*** : *Let the accuracy of Trojan model on data with embedded Trojan triggers to be* $(1 - \delta)$, *where* $\delta << 1$, *and let there be C different classes. If* $\Delta x_{rev}$ *is a Trojan trigger then the entropy computed by Eq 5 will be bounded by*

$$\mathrm{H}(\Delta x_{rev}) \leq -(1 - \delta) * \log_2(1 - \delta) - \delta * \log_2(\frac{\delta}{C - 1}). \tag{6}$$

The above lemma can easily be proved by observing that the highest entropy of class distribution in this setting happens when $(1 - \delta)$ fraction of the images go to the target class $t'$ and the rest $\delta$ fraction of the images gets equally distributed in the remaining $(C - 1)$ classes. This entropy score is independent of the type and size of triggers used and is universally applicable. This threshold computation has been adopted from STS Harikumar et al. (2021).

## 3 Experiments

We evaluate our proposed defense method on four datasets namely, MNIST, German Traffic Sign Recognition Benchmark (GTSRB) Stallkamp et al. (2011), CIFAR-10 Krizhevsky (2009), and YouTube Face Recognition (YTF) dataset Ferrari et al. (2018). We use Pre-activation Resnet-18 Nguyen & Tran (2020a) as the classifier for CIFAR-10 and GTSRB and Resnet-18 He et al. (2016) for YTF. We use a simple network architecture Nguyen & Tran (2020a) for MNIST dataset.

We train the Pure and Trojan classifiers using SGD Bottou (2012) with initial learning rate of 0.1 and used a learning rate scheduler after 100, 150, and 200 epochs, weight decay of 5e-4 and the momentum of 0.9. We use a batch size of 128 for CIFAR-10 and MNIST, and 256 for GTSRB with the number of epochs as 250. For YTF we use the batch size of 128 and the number of epochs as 50. For Trojan models, the target and non-target class ratio we have used is 70:30 ratio except for YTF which is 30:70 as it contains lots of classes and we found it hard to obtain a good pure accuracy with 70:30 poisoning ratio. While training the Trojan model, the ratio of Trojan data in a batch for MNIST and CIFAR-10 is set to 10% of the batch size 2% for GTSRB, and 0.2% for YTF. We have done all our experiments on DGX servers with 16 gpus.

We use square triggers of sizes 4×4, and 8×8 with trigger transparency of 1.0 to train MTTA model. In addition, we have used a 16×1 trigger to train a MTTA model to show the effectiveness of the defense in capturing the Trojan in the presence of a long and different size trigger. We have used a mask of size $H \times W$

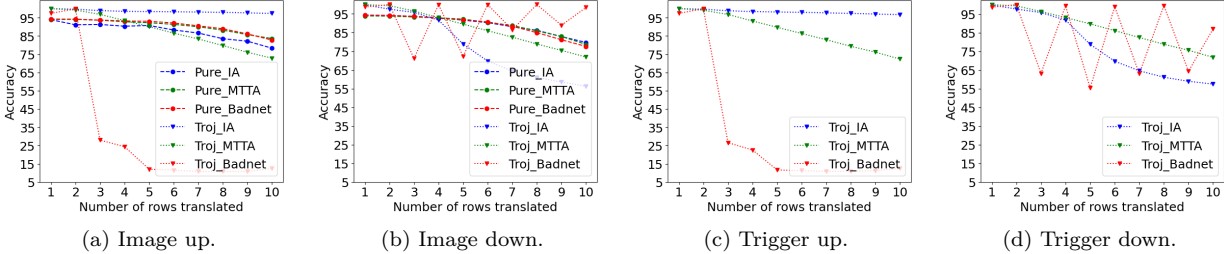

(a) Image up.      (b) Image down.      (c) Trigger up.      (d) Trigger down.

Figure 3: Robustness of MTTA attack against image/trigger translations. Pure and Trojan (denoted as Troj\_) accuracies vs the number of rows translated for Badnet, Input-aware attack (IA), and MTTA. Figure 3a shows the accuracy when we translate the images up, Figure 3b when we translate the images down, Figure 3c when we translate the triggers up, and Figure 3d when we translate the triggers down. Pure accuracies are not affected by trigger translations, and thus not reported.

with L1-regularization to show the effectiveness of our proposed MTD. We use random pixel values to create class-specific triggers. The purpose of random colored triggers is two-fold: a) *to show that the attack is potent even when triggers are not optimally distinct*, and b) *that the defense works without any structure in the triggers*. We use Adam optimizer Kingma & Ba (2015) with a constant learning rate of 0.01 for trigger reverse engineering. The details of the datasets, network architecture, and number of target classes for MTTA are reported in Table 1.

We term the accuracy computed on the pure data on the ground-truth labels as the *pure accuracy* and the accuracy on the Trojan data corresponding to the intended target classes as the *Trojan accuracy*. To demonstrate the strengths of MTTA, we also analyse two additional properties: a) **Robustness** - *how badly the Trojan accuracy is affected by* i) image translation, to mimic the misplacement of the object detection bounding box and ii) trigger translations to mimic the misplacement during physical overlaying of the trigger on the image. In both the cases a part of the trigger may get lost; and b) **Invisibility** - *how well Trojan data can hide from the pure classifiers*. If an attack is visible, then it would cause unintended side-effects by attacking pure classifiers too and thus compromise their stealth. We believe that strong robustness and complete invisibility are the hallmarks of an extremely potent Trojan attack.

We use six state-of-the-art defense strategies, i.e. STRIP Gao et al. (2019), Grad-CAM Selvaraju et al. (2017), FinePruning Liu et al. (2018), STS Harikumar et al. (2021), NAD Li et al. (2021) and Neural Cleanse Wang et al. (2019) to compare the proposed defense. However, STRIP and Grad-CAM are testing time defense and thus do not admit the same metric as Neural Cleanse, STS, and MTD. We do not compare with DeepInspect Chen et al. (2019) or GangSweep Zhu et al. (2020) as we believe that it would be unreasonable to train a GAN (used in both) with the small number of trigger reverse-engineering that we will be performing (20 -128) for different datasets.

## 3.1 Effectiveness of MTTA

The accuracies of the pure and the MTTA Trojan models are reported in Table 1. The performance shows that across various configuration choices, the proposed attack strategy succeeds in providing a high Trojan effectiveness (∼100%) whilst keeping the pure accuracy close to the pure model accuracy.

## 3.2 Robustness of MTTA to View Change

We use the MTTA Trojan model which was trained on CIFAR-10 with 8×8 Trojan triggers to demonstrate the robustness of MTTA under both slight misplacement of the image window and the trigger placement. To carry out the test, we translate the image up and down and pad the added rows (to match the original image size) with white pixels. For trigger translation we do the same just for the trigger before compositing it with the untranslated images.

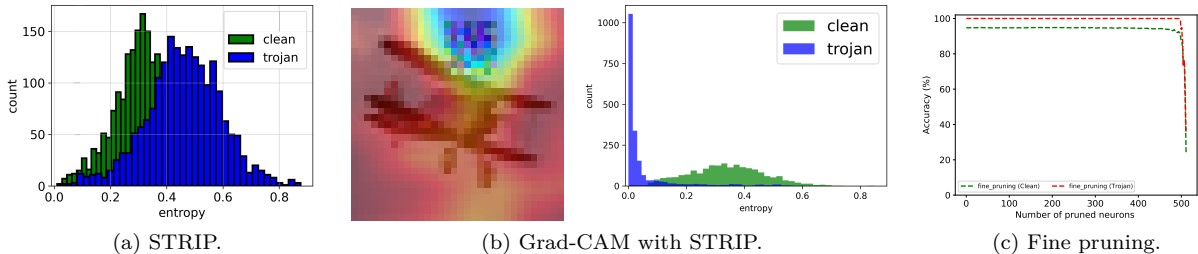

(a) STRIP.        (b) Grad-CAM with STRIP.        (c) Fine pruning.

Figure 4: a) STRIP results of CIFAR-10 on pure and Trojan data on a target class (class *6*), b) Grad-CAM and STRIP, c) Fine pruning on MTTA CIFAR-10 8×8 Trojan model.

| Pure data acc. (for reference) | Pure model acc. on Trojan data | | | | |
|---|---|---|---|---|---|
| | MTTA | | Badnet | | IA |
| | 4×4 | 8×8 | 4×4 | 8×8 | |
| 94.55 | 94.54 | 94.54 | 94.54 | 94.54 | 93.41 |

Table 2: Pure accuracy for Trojan data by a Pure model on both 4×4 and 8×8 triggers of CIFAR-10 dataset. The lowest accuracy is for the Input-aware attack.

The plots show the pure and Trojan accuracy of the Trojan models when we translate image up (Figure 3a), translate image down (Figure 3b), translate trigger up (Figure 3c), and translate trigger down (Figure 3d). We have chosen three attack models, the 8×8 trigger trained CIFAR-10 MTTA model, Input-aware attack (denoted as IA in Figure 3) and a Badnet trained with a 8×8 checkerboard trigger placed at the top-right corner. When translating up, we see that Badnet is disproportionately affected, whilst input-aware attack remained largely resilient. MTTA also dropped, but only slightly. When translating down, the Badnet remained resilient as the trigger patch remained within the translated image, but the Trojan effectiveness (Trojan accuracies) of input-aware attack dropped way more than MTTA.

When trigger is translated, the image underneath is not affected, and hence pure accuracies are not affected. But the Trojan effectiveness drops. However, we see again that whilst MTTA remains more or less resilient, in one case Badnet dropped catastrophically (Fig 3c) and in another case input-aware attack dropped way more than MTTA (Figure 3d). This shows that MTTA is a robust attack, especially when carried out in the physical space. The set of Figures from 3a - 3d shows the failure of the performance of the different existing attacks, however our approach maintains a consistency in performance throughout different settings.

### 3.3 Robustness of MTTA against test-time and Trojan mitigation defense

We have tested CIFAR-10 8×8 trigger (7 target classes (class *6, 9, 0, 2, 4, 3,* and *5*) with 7 different triggers for each target class) trained MTTA Trojan model against a state-of-the-art *test time defense* mechanisms such as STRIP Gao et al. (2019), Grad-CAM Selvaraju et al. (2017), and *Trojan mitigation mechanisms* such as Fine-pruning Liu et al. (2018), and NAD Li et al. (2021). STRIP, Grad-CAM, Fine-pruning and NAD are different kinds of remedial measures that try to use the infected model, without needing to ever detect one being a Trojan model. Here we show that MTTA is resistant to all these remedial measures.

#### 3.3.1 Test-time defense

Test time defense are useful in screening the inputs. Figure 4a shows the entropy plots of pure images and Trojan images for a target class (class *6*) after performing STRIP. The threshold of the entropy is calculated from the pure images by assuming that it follows a normal distribution. The 1% of the normal distribution of the entropy of the pure images will be chosen as the threshold to separate pure and Trojan images. So, during test time, any inputs which have an entropy value above the threshold will be considered as a pure image. The False Positive Rate (FPR) and the False Negative Rate (FNR) based on 1% threshold is 0.02% and 99.40% respectively. From the Figure 4a, it is evident that it is difficult to separate the pure and Trojan

| Dataset | Target classes |
|---------|----------------|
| MNIST | **6, 9, 0, 2, 4, 3, 5** |
| GTSRB | **24, 26, 2, 16, 32, 31, 25, 19,29, 34, 27, 40, 10, 15, 28, 1 30, 22, 13, 8, 9, 4, 42, 3,35, 36, 37, 17, 33, 41** |
| CIFAR10 | **6, 9, 0, 2, 4, 3, 5** |
| YTF | 30% Trojan (**384** target classes) and 70% Pure (**899** pure classes) among **1283** classes. |

Table 3: List of target classes of MNIST, GTSRB, CIFAR10, and YTF.

images with the computed threshold (shown as a red dotted vertical line). The results show that STRIP totally fails to defend against MTTA attacks.

Grad-CAM Selvaraju et al. (2017) introduced a mechanism to provide explanations of images by utilizing gradient information. We have used Grad-CAM to inspect an input with a trigger as shown in Figure 4b. These explanations are extracted and later used to detect the presence of a trigger in input during test time with STRIP Gao et al. (2019). We note that the combination of Grad-CAM and STRIP has not been previously done. The heat map of the Trojan image in Figure 4b shows that the Grad-CAM is picking up the trigger region. However, the results vary greatly depending upon which layer is chosen for Grad-CAM inspection. In our experiments, we have found that layer 3 is the best. We have converted the heatmap from Grad-CAM to a mask using a threshold of 0.5. This mask will later be used to blend the Trojan image with the set of pure images. It is clear from Figure 4b that there is an overlap exists between the pure and the Trojan entropy distribution. It is more successful compared to STRIP; however, it still cannot stop the Trojan images without considerable drop in efectiveness. For example, if we allow only 1% of pure images to be Trojan the False Negative Rate of Trojan images as pure is 30.35%.

### 3.3.2 Trojan mitigation

Fine Pruning Liu et al. (2018) results of the CIFAR-10 $8 \times 8$ trigger trained MTTA model is shown in Figure 4c. This mechanism removes the least activated neurons based on the pure images the defender has access to. Thus, this mechanism will prune the neurons which are highly influenced by the Trojan features and drop the Trojan accuracy of the model. It is clear that the Trojan and pure accuracy remains intact as the pruning progresses and drops together only when the number of pruned neurons is equal to the total number of neurons - something that is not at all desirable.

Neural Attention Distillation Li et al. (2021) uses attention-based knowledge distillation to finetune a student model from the given Trojan model. We have done experiments on results on the CIFAR-10 $8 \times 8$ trigger trained MTTA model which drops pure accuracy by $\sim 5\%$, and Trojan accuracy by $\sim 45\%$. Although somewhat reduced, NAD was unable to remove the Trojan effect totally.

### 3.4 Invisibility of MTTA

Here, we test the invisibility of the MTTA attack against pure models. We use a pure model to check for the pure accuracy for images under all three attacks: MTTA, Badnet, and input-aware attack. It is clear from the Table 2 that both the Badnet and MTTA attacks are invisible to the pure model. However, there is a slight drop ($>1\%$) of performance for input-aware attacks. This is expected because the amount of changes for input-aware (IA) attacks is much higher than MTTA and Badnet. Even a slight drop in pure accuracy might be enough to make it vulnerable to early detection.

| Dataset | Original classes | Predicted classes | Intrinsic Trojan classes |
|---------|------------------|-------------------|--------------------------|
| GTSRB | **1, 2, 3, 4, 8,9, 10, 13, 15, 16, 17, 19, 22, 24, 25, 26, 27, 28, 29, 30, 31, 32, 33, 34, 35, 36, 37, 40, 41, 42** | **0, 1, 2, 3, 4, 5, 6, 7, 8, 9, 10, 11,12, 13, 14, 16, 17, 18, 19, 20, 21, 22, 23, 24, 25, 26, 27, 28, 29, 30, 31, 32, 33, 34,35, 36, 37,38, 39, 40, 41, 42** | **0, 5, 6, 7, 38, 39** |

Table 4: Original and predicted target classes of GTSRB $4 \times 4$ triggers trained MTTA model. The intrinsic Trojan classes are non-target classes but are predicted as target classes.

| Dataset | F1-score model detection | | |
|---------|-----|-----|-----|
| | NC | STS | MTD |
| MNIST | 0.0 | 0.8 | **1.0** |
| GTSRB | 0.0 | 1.0 | **1.0** |
| CIFAR-10 | 0.0 | 0.8 | **1.0** |
| YTF | 0.0 | 1.0 | **1.0** |

(a) Model detection.

| Dataset | F1-score target class detection | | | |
|---------|-----|-----|-----|-----|
| | 4×4 | | 8×8 | |
| | NC | MTD | NC | MTD |
| MNIST | 0.0 | **0.92** | 0.0 | **1.0** |
| GTSRB | 0.0 | **0.81** | 0.0 | **0.77** |
| CIFAR-10 | 0.0 | **1.0** | 0.0 | **1.0** |
| YTF | 0.0 | **0.43** | 0.0 | **0.46** |

(b) Target class detection.

Table 5: a) F1-score for MTTA Trojan model detection using MTD (90% Trojan effectiveness for entropy threshold computation), NC (anomaly detection threshold 2.0), and STS. b) F1-score for detecting the target class in MTTA based on our proposed defense MTD.

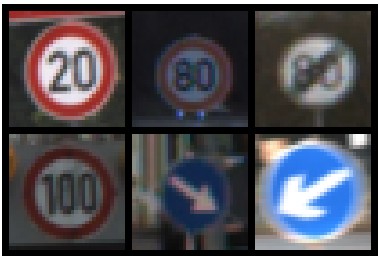

Figure 5: The non-target classes of GTSRB dataset which have intrinsic Trojans in them are class *0 (Speed limit 20 km/hr), 5 (Speed limit 80 km/hr), 6 (Speed limit 80 km/hr end), 7 (Speed limit 100 km/hr), 38 (Keep right),* and *39 (Keep left).*

### 3.5 Intrinsic Trojans in non-target classes

Intrinsic Trojans are the properties of the non-target classes which are similar to the target classes of a MTTA model. The introduction of triggers in target classes can produce intrinsic Trojan patterns in non-target classes, mainly because of the feature similarity between the target and non-target classes. This can result in the detection of non-target classes as target classes. For example, in GTSRB, class *0 (Speed limit 20 km/hr), 5 (Speed limit 80 km/hr), 6 (Speed limit 80 km/hr end),* and *7 (Speed limit 100 km/hr),* are detected as target classes, this can be because of the Trojans in class *1 (Speed limit 30 km/hr), 2 (Speed limit 50 km/hr), 3 (Speed limit 60 km/hr), 4 (Speed limit 70 km/hr),* and *8 (Speed limit 120 km/hr).* Similarly, class *38 (Keep right),* and *39 (Keep left)* are also detected as target classes due to the presence of Trojans in class *33 (Turn right), 34 (Turn left), 35 (Only straight), 36 (Only straight or right),* and *37 (Only straight or left).*

Table 3 reports the target classes (Trojan inserted classes can be called as Trojan classes) for all the datasets MNIST, German Traffic Sign Recognition Benchmark (GTSRB) Stallkamp et al. (2011), CIFAR10 Krizhevsky (2009), and YouTube Face Recognition (YTF) dataset Ferrari et al. (2018) for reference. We

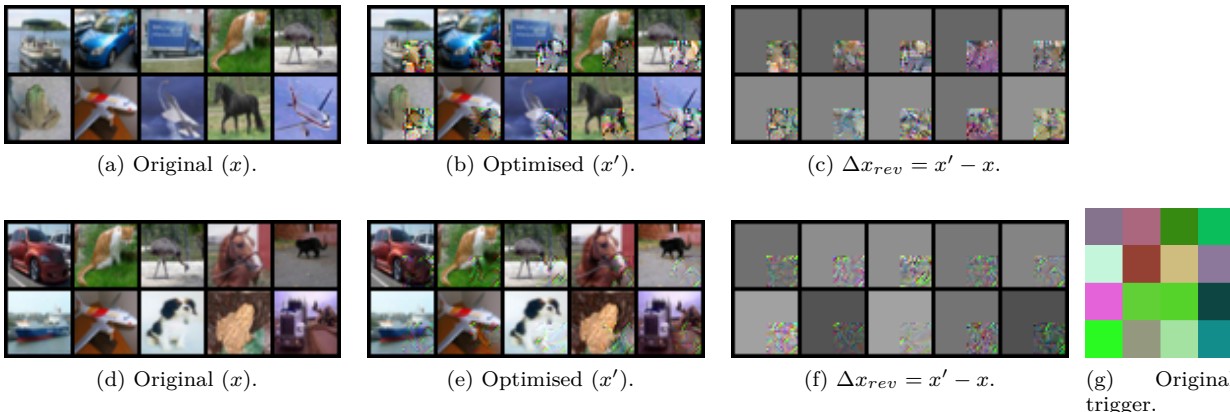

(a) Original ($x$).     (b) Optimised ($x'$).     (c) $\Delta x_{rev} = x' - x$.

(d) Original ($x$).     (e) Optimised ($x'$).     (f) $\Delta x_{rev} = x' - x$.     (g)     Original trigger.

Figure 6: Sample reverse-engineered triggers for non-Trojan class (top row) and Trojan class (bottom row) of a CIFAR-10 Trojan model with $4 \times 4$ trigger. Please zoom in to see how the non-Trojan perturbations are optimising more towards part of the image of the target class whilst the Trojan triggers are optimising towards the original trigger.

use the 7:3 ratio for target and non-target classes except for YTF. For YTF we use a 3:7 ratio for target and non-target classes. Table 4 reports the original and predicted target classes of GTSRB $4 \times 4$ trigger trained MTTA model. The predicted target classes are computed from the entropy of the reverse-engineered triggers. It is evident that some non-target classes are being predicted as target classes. The classes which are originally non-target classes but have got intrinsic Trojans are also reported (third column). The images of the set of non-target classes (class *0 (Speed limit 20 km/hr), 5 (Speed limit 80 km/hr), 6 (Speed limit 80 km/hr end), 7 (Speed limit 100 km/hr), 38 (Keep right),* and *39 (Keep left)* which has got intrinsic Trojans in it are shown in Figure 5.

### 3.6 Trojan Detection by our proposed MTD

We look at both the class-wise detection and model-level detection performance. A class is declared Trojan if any of the recovered triggers for that class produces a class-distribution entropy that is lower than the threshold (as per Eq 6). We use $\delta = 0.1$ in all our experiments. A model is flagged as Trojan when at least one of the classes is Trojan.

#### 3.6.1 Model Detection

We have a pure model and two Trojan models corresponding to two different trigger sizes for each dataset. The F1-score of the model detection by Neural Cleanse (NC), STS, and our method (MTD) is shown in Table 5a. It is clear from the Table that NC failed to detect Trojan in the MTTA setting. It is also interesting to note that Pure models of all the datasets are getting detected as Trojan models in NC. Though STS perfectly detected Trojan models as Trojan it detected pure model as Trojan in the case of CIFAR-10 and MNIST. STS has achieved an F1-score of 1.0 for GTSRB and YTF datasets. However, our proposed detection mechanism MTD has an F1-score of 1.0 for all the datasets clearly separating Trojans from pure.

#### 3.6.2 Class-wise Detection

In Table 5b we report the F1-score of NC and MTD in detecting the target classes. We didn't report results from STS since it can detect at most one Trojan class. For MNIST and CIFAR-10 it was able to detect the target classes correctly. However, for GTSRB and YTF, there has been a drop in the class-wise detection performance. The drop happens because in those datasets (traffic signs, and faces) many of the classes are quite similar and when among a group of similar classes if one is the target class then we observe that many of the others also happen to be detected as target class as well. This is expected because the shortcut

| Dataset | F1-score | | | | | | | | | |
| --- | --- | --- | --- | --- | --- | --- | --- | --- | --- | --- |
| | 4×4 trigger | | | | | 8×8 trigger | | | | |
| δ | *0.01* | *0.05* | *0.1* | *0.15* | *0.20* | *0.01* | *0.05* | *0.1* | *0.15* | *0.20* |
| MNIST | **0.92** | 0.82 | 0.82 | 0.82 | 0.82 | **1.0** | 1.0 | 0.93 | 0.93 | 0.93 |
| GTSRB | 0.28 | 0.66 | **0.81** | 0.82 | 0.82 | 0.56 | 0.61 | **0.77** | 0.82 | 0.82 |
| CIFAR-10 | **1.0** | 1.0 | 1.0 | 1.0 | 0.93 | **1.0** | 1.0 | 1.0 | 1.0 | 0.93 |
| YTF | 0.02 | 0.28 | **0.43** | 0.46 | 0.45 | 0.12 | 0.37 | **0.46** | 0.45 | 0.45 |

Table 6: The performance of our proposed defense MTD with respect to $\delta$ on MTTA Trojan models trained on different datasets.

introduced by a target class also end up serving the classes close by. As expected, NC failed badly because it was not designed to detect multi-target attack.

The initial set of images, the optimized images, the difference between the given images and the optimised images ($\Delta x_{rev}$), of a non-target class (*class 7*) and a target class (*class 6*) is shown in the top and bottom rows of Figure 6, respectively for a CIFAR-10 Trojan model trained with $4 \times 4$ trigger. The $\Delta x_{rev}$ of the non-Trojan class samples have no visible trigger patterns in it, however, for the Trojan class there are some patterns which look like the original trigger as shown in Figure 6g.

### 3.7 Ablation Study

#### 3.7.1 Performance Vs $\delta$

We report the F1-score of Trojan class detection for Trojan models based on different values of $\delta$. The results show that as we increase $\delta$, the F1-score reduces. This is because with higher $\delta$ many non-Trojan classes are also classified as Trojan classes. We find that $\delta = 0.1$ provides the most stable results across all the datasets.

#### 3.7.2 Performance with, without and full size mask ($H \times W$)

We use the 8×8 triggers trained CIFAR-10 MTTA model to perform experiments with and without a mask. Our method won't be able to detect the Trojan classes and hence the Trojan model for without mask. However, it achieves a perfect F1-score (1.0) when we use mask as shown in Figure 7b.

We use a $16 \times 1$ trigger-trained CIFAR-10 MTTA model to perform experiments with the size of the mask as same size of the input image ( a sample is shown in Figure 7c). We use this trigger which is quite long to show the effectiveness of the proposed method in situation where we don't assume any pre-assumption of the trigger size. We have used L1-regularization to control the sparsity of the perturbations. The F1-score for detecting the MTTA model and the target classes are shown in Figure 7b. This shows the utility of our proposed defense method against MTTA models trained with triggers of various shapes.

#### 3.7.3 Single target attack

We choose a Badnet trained on CIFAR-10 dataset with $4 \times 4$ trigger and apply our MTD. For the Badnet, *class 0* is the target class, and the rest are non-target classes. When MTD is applied, only the target class is detected as Trojan and all the non-target classes are detected as non-Trojan. The entropy plots which are shown in Figure 7a of the Trojan (*class 0*) and a randomly sampled a non-Trojan class (*class 2*) demonstrate the difference between the entropy distributions.

#### 3.7.4 Robustness against adaptive attack

What if the attacker knows about our defense mechanism? The attacker can do one of the two things: a) can use a large trigger or b) can use a trigger that only works on a portion of the dataset such that the transferability property is invalid. The first one is essentially coming at a cost of decreased stealth and thus may not be feasible. The second one does not cost anything, but it requires that we can divide the data

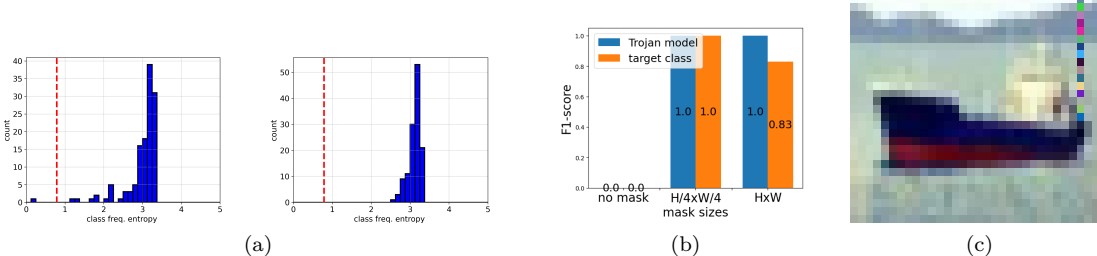

Figure 7: a) Distribution of class distribution entropies computed over many recovered triggers for both Trojan (left) and non-Trojan class (right) for single target Badnet attack. Only the Trojan class has some triggers that resulted in entropy scores lower than the threshold (red dashed line) b) F1-score of Trojan model and target class detection (with 90% Trojan effectiveness) with different mask sizes, c) a sample of 16×1 trigger.

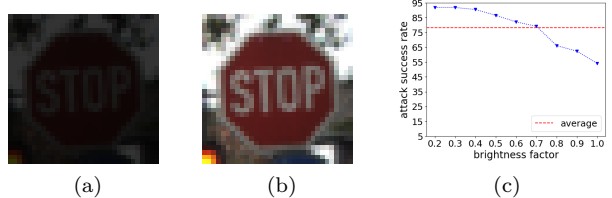

Figure 8: a) Training image (cloudy day) for adaptive attack, b) same image (test image) on a sunny day for adaptive attack, c) Adaptive Attack with different brightness factors.

distribution of a class into two or more non-overlapping sub-distributions. However, we think such a division would be very unlikely to be achieved for real data. We have simulated the above-discussed settings using the GTSRB dataset. We have created two sub-distributions within a selected class (STOP sign) by making one set of images darker and the other set brighter. The darker images resemble images that are taken on a cloudy day. The other set resembles the images that are taken on sunny day. We train the model with only the darker set of images. This trained Trojan model was later tested on images from sunny days. The sample image used for training and testing for the adaptive attack is shown in Figure 8a and 8a. On average, the Trojan attack success rate is 78% with the new distribution of images as shown in Figure 8c. So, any potential trigger will be still transfer to a large portion of the *Data_Transfer* set (78%).

## 4 Conclusion

In this paper, we proposed a variation of the Badnet style attack on multiple targets that can defeat six state-of-the-art defense mechanisms and are more robust than recent attacks. We then proposed a new detection method based on reverse-engineering triggers for individual images and then verifying if a recovered trigger is transferable. We use a class-distribution based entropy mechanism to compute a threshold that would separate the Trojan triggers from the rest. Our extensive experiments with four image datasets of a varying number of classes and sizes show that we can classify pure and Trojan models with a perfect score. We tested the efficacy of the attacks on fixed image datasets only. In physical domain our proposed attack may become less robust due to the presence of environmental disturbances. However, that will affect all attack methods. In relative terms, MTTA may still provide a better attack model especially when the attack is carried out in physical space. When environmental conditions are more favorable e.g., in clear daylight, Trojan attacks would work quite well and thus they still pose a significant threat. A future possible study is to investigate the cases when a pure detection dataset is not available or when purity cannot be guaranteed.

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
