# OpenReview forum: "Defense Against Multi-target Backdoor Attacks"
_TMLR — Rejected by TMLR_

### Review · Reviewer_NxLu · 2024-04-21

**Summary Of Contributions:**

This paper proposes a Trojan attack and provides a corresponding defense method. The prosed attack uses multiple triggers to control multiple target classes, and does not restrict the location of the trigger in the image. Since existing defense methods fail to detect the proposed attack, the authors further provide mitigation for this attack.

**Audience:**

Yes

**Claims And Evidence:**

Yes

**Requested Changes:**

Please conduct a comprehensive literature review for recent studies and consider some recent methods for comparison.

**Strengths And Weaknesses:**

This paper provides a lot of experiments demonstrating the effectiveness of their proposed methods.

However, there are some weaknesses:

(1) Among the 29 cited papers, only one of them is published after 2021. The authors are suggested to conduct a more comprehensive literature review to highlight their difference with the other recent literature.

(2) The experiment is not sufficient. First, there is no adequate comparison with other recent methods as the paper does not cite any recent paper. Second, the authors are also suggested to consider experiments in vision transformer and ImageNet.

(3) While the motivation of this paper is clear, it lacks descriptions about the novelty of the proposed method. It is not clear the difficulties when designing/implementing the proposed algorithm. There is also insufficient insights on why the proposed algorithm can be robust.

(4) From my understanding, the trigger size can be affected by the image size. The authors are suggested to perform experiments on data with higher resolutions (ImageNet and other datasets with even higher resolutions).

---

> ### Author Response · Authors · 2024-05-14
> **Response to Reviewer NxLu**
>
> We thank the reviewer for the valuable comments. Please see our responses below:
>
> 1. Comprehensive literature review to highlight their differences with the other recent literature.
>
> We understand the concern raised by the reviewer. Based on the suggestion we conducted a thorough analysis and compared our work with some recent state-of-the-art backdoor mechanisms and reported the results (Please refer to the response to Reviewer 4Cqb). We will incorporate all these details and cite all the recent and relevant papers in the main paper.
>
> 2. The experiment is not sufficient. Second, the authors are also suggested to consider experiments in vision transformer and ImageNet.
>
> We did studies on a total of five datasets which are high to low resolutions and four different model architectures for the paper. For the rebuttal, we did additional experiments with the **TinyImagenet** dataset (a modified subset of the original ImageNet dataset) that has a size of image **64x64x3** with **200** classes as ImageNet dataset experimentation can be time-consuming given the period of rebuttal submission. Our proposed MTD detects the pure and Trojan models of the TinyImageNet dataset with **1.0**  F1-score. However, we have only got **0.70** F1-score for the target class detection of the TinyImagenet 8x8 MTTA model.  We also did some analysis based on the VGG11 on the Cifar10 dataset. We also compare our work with the recent state-of-the-art model detection approach FreeEagle[1], ANP[2], and RNP[3]. Please refer to the response to Reviewer 4Cqb for the analysis of the results.
>
> 3. Insufficient insights on why the proposed algorithm can be robust.
>
>  By robustness what we meant is the potential of MTTA (attack) to be physically applicable in the presence of various environmental factors (misplacement of object detection bounding box), misplacement of triggers, sunny or rainy days (Figure 8(a,b,c) in main paper).
>
> 4. From my understanding, the trigger size can be affected by the image size. The authors are suggested to perform experiments on data with higher resolutions.
>
> We conducted additional experiments with **TinyImagenet** (a modified subset of the original ImageNet dataset) dataset which has a size of 64x64x3 and has 200 classes. We have a pure and 8x8 trigger MTTA model with 60 target classes. and have shown that our proposed method detects pure and MTTA models successfully. Our proposed MTD detects the pure and Trojan models trained by Tiny ImageNet with **1.0**  F1-score. However, we have only got **0.70** F1-score for the target class detection by the proposed MTD. Additionally, the analysis of the YouTube face recognition dataset models (results reported in Table 5(a,b) in the main paper) also shows the potential of our method when we have high-resolution images.
>
> [1] FreeEagle: Detecting Complex Neural Trojans in Data-Free Cases, Fu et al., USENIX Security 2023.
>
> [2] Adversarial Neuron Pruning Purifies Backdoored Deep Models, Wu et. al, NeurIPS 2021.
>
> [3] Reconstructive Neuron Pruning for Backdoor Defense, Li et al., ICML 2023.

---

### Review · Reviewer_4Cqb · 2024-04-28

**Summary Of Contributions:**

This paper considers a backdoor attack scenario where attackers use multiple triggers to control different classes. Then the paper proposes a backdoor detection framework based on trigger reserve-engineering. The experimental results demonstrate the superiority of the framework against other methods.

**Audience:**

Yes

**Broader Impact Concerns:**

No Concerns

**Claims And Evidence:**

Yes

**Requested Changes:**

-	I recommend the authors to address the weakness above. The most important is to consider stronger baselines into experiment evaluation.

**Strengths And Weaknesses:**

Strengths:
-	The backdoor scenarios considered in this paper allow attackers to use different triggers for each class and attack multiple categories, which is more practical.
-	The experimental results on various datasets demonstrate advantages in terms of model detection and target class detection.
-	This paper does a good job of conveying its main technical ideas.
Weaknesses:
-	My main concern arises from the chosen baseline. As far as I understand, backdoor detection/defense methods like Neural Cleanse, STRIP, and NAD are not currently the strongest methods. When facing backdoor attacks that are stronger than vanilla Badnets, these methods indeed sometimes perform poorly. I believe there are stronger detection/defense frameworks in the AI/security community, such as MNTD[1], FreeEagle[2] for detection, and ANP[3], BNP[4] for mitigation, and so on.
-	I would like to know how the proposed method performs on other models such as VGG.
[1] Xu X, Wang Q, Li H, et al. Detecting ai trojans using meta neural analysis.
[2] Shen G, Liu Y, Tao G, et al. Backdoor scanning for deep neural networks through k-arm optimization.
[3] Wu D, Wang Y. Adversarial neuron pruning purifies backdoored deep models.
[4] Zheng R, Tang R, Li J, et al. Pre-activation distributions expose backdoor neurons.

---

> ### Author Response · Authors · 2024-05-14
> **Response to Reviewer 4Cqb**
>
> We thank the reviewer for the valuable comments. Please see our responses below:
>
> 1. Stronger detection/defense frameworks in the AI/security community.
>
> We understand the concerns and we compared our proposed attack with the recommended state-of-the-art model detection mechanisms such as FreeEagle[1] and model mitigation mechanisms like ANP[2] and RNP[3]. We use CIFAR-10 pure, 4x4 trigger, and 8x8 trigger MTTA models for the experiments.
>
> **Table 1**
> | CIFAR10 | F1-score model detection (FreeEagle) |
> | --------     | -------:|
> | 1.5 anomaly threshold  | 1.0   |
> | 2.0 anomaly threshold  | 0.6  |
>
> We reported the F1 scores of model detection under different anomaly thresholds in Table 1. The anomaly scores for the 4x4 Trojan model and pure cifar10 models by FreeEagle are very close, **1.77** and **1.09**, respectively. We have also noticed that several classes in Pure models are also being detected as target classes (based on the score we got for each class).
>
> **Table 2**
> | CIFAR10 | F1-score target class detection (FreeEagle) |
> | --------     | -------:|
> | 4x4 MTTA | 0.22   |
> | 8x8 MTTA | 0.44  |
>
> We report the F1-score of the FreeEagle in detecting target classes of the given 4x4 and 8x8 MTTA (Trojan) models in Table 2. The performance of FreeEagle is not as effective as compared to our proposed MTD detection mechanism which achieves an F1-score of 1.0 for both 4x4 and 8x8 MTTA models (main paper Table 5, page number 9).
>
> **Table 3**
> | Defense |Accuracy | 35 |70 |105|140 |175 |210 |245 |280 |315 |
> | -------- | ---- | ---- |---- |---- |---- |---- |---- |---- |---- |---- |
> | ANP| Pure|94.31 |94.06 |93.98 |93.74|93.46|93.48 |92.93|92.72|92.60|
> | ANP| Trojan|97.33|83.73|79.07 |70.91|36.83|32.90|24.08|24.06|21.23|
> | RNP| Pure|89.27 |86.44|78.42 |76.98|70.71|57.20|56.80|51.35|40.40|
> | RNP| Trojan|27.23|19.29|11.22|12.13|11.19|10.96|12.07|11.65|11.58|
>
> We reported the results of ANP and RNP mitigation approaches in Table 3. We used our MTTA 8x8 Cifar10 model to conduct the experiments. The columns of Table 3 represent the number of pruned neurons during the Trojan mitigation processes. The pure and the Trojan accuracies are the accuracies of the MTTA model to the number of pruned neurons. It is clear from the Tables that it is hard to completely remove the signature of triggers from the proposed MTTA model without compromising pure accuracy as the pruning progresses.
>
> **Table 4**
> | Defense |Accuracy | 35 |70 |105|140 |175 |210 |245 |280 |315 |
> | -------- | ---- | ---- |---- |---- |---- |---- |---- |---- |---- |---- |
> | ANP| Pure model acc. |94.45|94.50|94.47|94.19|94.03|93.95|93.70|93.59|92.62|
> | RNP| Pure model acc.|91.23|90.21|87.61|86.35|84.66|83.28|79.67|77.62|74.53|
>
> Table 4 shows the accuracy of a given pure cifar10 model when we prune the neurons based on ANP and RNP. From Table 4, it is clear that both ANP and RNP do pruning no matter whether it is a Trojan or pure model. An unnecessary reduction in the accuracy of pure models is still a major concern. So, it is still important to detect whether the given model is pure or Trojan and its corresponding target classes.
>
> 2. The proposed method performs on other models such as VGG.
>
> We conducted experiments with the CIFAR10 dataset on VGG11. We have a pure VGG11 model and two MTTA VGG11 models trained with 4x4 and 8x8 triggers respectively. The target classes and other settings are similar to what we have discussed in the experiment Section 3.
>
> **Table 5**
> | Dataset| F1-score model detection (MTD (ours) |
> | --------     | ------: |
> | Cifar10| 1.0|
>
> The Trojan model detection performance of our proposed MTD is shown in Table 5. It has 100% success in distinguishing the pure and the Trojan VGG11 models.
>
> **Table 6**
> | Model| F1-score target class detection (MTD (ours) |
> | --------     | -------: |
> | 4x4 MTTA cifar10| 0.83|
> | 8x8 MTTA cifar10| 1.0|
>
> We also identified the target classes of the 4x4 and 8x8 MTTA models using MTD as shown in Table 6. The values in Table 6 show that MTD has 100% success in detecting all the target classes of the 8x8 MTTA model, however, it dropped for the 4x4 trigger MTTA to 83%. Additionally, we already have a result for a different and smaller CNN architecture (Table 1 in the main paper) MNIST dataset. The results are reported in Table 5(a,b) in the main paper.
>
> 3. Comprehensive literature review for recent studies.
>
> We acknowledge that our literature review is not extensive. However, following the reviewer's suggestion, we conducted a detailed analysis of the recent studies to compare our proposed attack and defense mechanisms. We will incorporate all these details in the main paper.
>
> [1] FreeEagle: Detecting Complex Neural Trojans in Data-Free Cases, Fu et al., USENIX Security 2023.
>
> [2] Adversarial Neuron Pruning Purifies Backdoored Deep Models, Wu et. al, NeurIPS 2021.
>
> [3] Reconstructive Neuron Pruning for Backdoor Defense, Li et al., ICML 2023.

---

### Review · Reviewer_bywn · 2024-05-01

**Summary Of Contributions:**

- The paper introduces a variant of Badnet that utilizes multiple triggers to control multiple target classes. Moreover, these triggers can be placed at any location within the input image, making the attack more potent and practical for real-world scenarios.
- The authors empirically demonstrate that many existing defenses against Trojan attacks are ineffective against the proposed variant.
- The proposed attack and defense mechanisms are evaluated on various image datasets.

**Audience:**

Yes

**Claims And Evidence:**

Yes

**Requested Changes:**

There are many technical parts need more justification, which hinder the understanding of the paper. I list the points below.
1. Abstract:
 "In this paper, we first introduce a variant of BadNet that uses ... "
What is Badnet? This pop-up make readers confusing.

2. Page 2:
"The intuition of our method is illustrated in Figure 1 through an understanding of the classification surface in pure and Trojan models."
Where does this understanding comes from? Is it supported by any reference? If it is just an assumption or conjecture, the argument is weak.

3. Page 2:
" A Trojan trigger would cause most of the images to go to the same class, ..."
It is confusing that the trigger would lead to same class. Also, I don't see how "multi-target" is developed in the introduction.

4. Page 4:
"If a perturbation is the Trojan trigger, then it will transfer to all the images and the class distribution would be peaky at the target class, resulting in a small entropy value."
Again, this point is confusing.

5. Table 1:
To me, these datasets and classifiers are kind of old and simple, have you tried your method on other foundation models and newer datasets?

6. Section 2:
I couldn't find how MTTA is developed. It seems like only MTD is introduced.

7. Lemma 1:
The lemma is not clear to me.

8. Section 3:
"b) Invisibility - how well Trojan data can hide from the pure classifiers."
How invisibility is quantitatively measured? Again, I didn't find how "multi-target" is addressed.

9. Section 3.2:
I don't understand the point of studying the robustness of MTTA.
Moreover, there are typos scattering around , e.g, "associated defense as Multi-Target Defense (MTD."

**Strengths And Weaknesses:**

Stengths:
+ The paper not only proposes new attack and defense mechanisms but also empirically evaluates them on various image datasets.
+ This approach adds a practical dimension to the defense against Trojan attacks, potentially mitigating their impact in real-world settings.

Weaknesses
- The generalizability of the experimental results to different domains or datasets with different characteristics could be further explored to assess the scalability and adaptability of the proposed approaches. The trial dataset is small and kind of outdated.
- I cannot find the description of MTTA, which is an important component of the proposed method.
- Also, it is not clear to me that how multi-target is addressed in the paper.
- There are many technical details confusing and require further explanation.

---

> ### Author Response · Authors · 2024-05-14
> **Response to Reviewer bywn**
>
> We thank the reviewer for the valuable comments. Please see our responses below:
>
> 1. The generalizability of the experimental results to different domains or datasets.
>
> We performed additional experiments with the TinyImagenet dataset which has a size of 64x64x3 and has 200 classes. We used a trigger of size 8x8 and selected 60 target classes for the MTTA model. The threat model, MTTA has a 100% attack success rate and our proposed MTD model detected the Trojan and the pure model successfully, for target class detection we got an F1-score of 0.70.
>
> 2. Also, it is not clear to me how multi-target is addressed in the paper.
>
> We have discussed this in Algorithm 1 on page 4 and also discussed it in detail in Section 2.2.
>
> 3. Abstract: "In this paper, we first introduce a variant of BadNet that uses ... " What is Badnet?
>
> We had the preassumption that the Badnet [Gu et al. 2017] is a very well-known attack and known to the backdoor attack community. We apologize for this, and we will remodify the section having in mind a wider set of readers.
>
> 4. Page 2: "The intuition of our method is illustrated in Figure 1"
>
> This is a conjecture. The basis for this is that since a trigger can effortlessly convert any image from its original class to the target class the trigger needs to be in an orthogonal feature subspace to the feature space of the original class. The success of our method indicates that our conjecture is true.
>
> 5. Page 2: " A Trojan trigger would cause most of the images to go to the same class, ..." It is confusing that the trigger would lead to the same class. Also, I don't see how "multi-target" is developed in the introduction.
>
> We kindly request the reviewer to refer to the threat model MTTA (sec 2.1). This provides the intention and results in (Table 1, col: Trojan accuracy) demonstrates that such is possible to achieve through appropriate training. We also note that this is a general concept behind the Backdoor insertion in the model, including the pioneering work by Gu et al. (Badnet).
>
> 6. Page 4: "If a perturbation is the Trojan trigger, then it will transfer to all the images and the class distribution would be peaky at the target class, resulting in a small entropy value."
>
> This is connected to the previous answer that if the revere-engineered pattern is indeed the original trigger then it would have the transferability because of the phenomenon mentioned in the previous answer.
>
> 7. Have you tried your method on other foundation models and newer datasets?
>
> Our method should also scale to the foundation model. However, the foundation models tend to come from more well-known developers and thus are not plagued by the same security issues. We feel the backdooring issue will be more prevalent on boutique, or custom models where the data supply chain is less secure.
>
> 8. Section 2: I couldn't find how MTTA and only MTD is introduced.
>
> The attack we propose in our paper is MTTA attack which is defined in the threat model section 2.1. We will adjust the title to reflect that. Thank you for pointing this out.
>
> 9. Lemma 1 not clear.
>
> We propose a universally applicable score to detect anomalous behaviour of the potential reverse-engineered triggers reported in Lemma 1. We say a model is Trojaned if any of the reverse-engineered triggers have an entropy below the score equal to or below the score computed based on Lemma 1. This is demonstrated in Figure 2 and Algorithm 1. The varying factor in the lemma score computation is trojan effectiveness, for most of our analysis we used 99% trojan effectiveness i.e  99% attack success rate.
>
> 10.	Section 3: "b) Invisibility - "
>
> We show in Table 2 that if we present the Trojan data (designed for the MTTA and Badnet models) to a pure model (a model that is not trained with Trojan trigger), it acts normally as the performance of any pure model should not be hampered by the presence of a trigger in the data.
>
> 11.	Section 3.2: I don't understand the point of studying the robustness of MTTA and typos.
>
> The robustness we conducted is essentially to demonstrate how the influence of external factors can affect the attack success rate and the stealth of  MTTA. It shows the potential of MTTA to be carried out in physical space. Thanks for pointing it out, we will correct all the concerned typos.

---

### Review · Reviewer_kYEf · 2024-05-01

**Summary Of Contributions:**

The authors propose a new Trojan attack method for images called multi-target Trojan attack which defeats many of the existing defense algorithms. However, the poor writing quality of the paper makes it difficult to evaluate their contributions.

**Audience:**

Yes

**Claims And Evidence:**

No

**Requested Changes:**

Please address the writing clarity issues and weaknesses listed above.

**Strengths And Weaknesses:**

- Although the authors perform a lot of experiments and analysis of their data and methods, the poor writing quality of the paper makes it very difficult to understand their contributions.

- The rationale behind Eqt 3 is not clear. Why do we want to run the classifier f_\theta over a 4x4 or 8x8 patch with the rest of the pixels zeroed out? Shouldn't we directly optimize over the contaminated image f_\theta(x + B(x)) instead?

- What are the definitions of pure accuracy and Trojan accuracy in Table 1? The accuracy under 4x4 and 8x8 triggers are very high. Does that mean that the Trojan attacks are not effective? There is a one sentence description of the terms at the beginning of Section 3 but they are not clear. Please put down rigorous definitions in terms of equations.

- Table 2 shows the proposed method MTTA performs essentially the same as previous methods like Badnet. What then is the benefit of the new method?

- In 2.2.2 for Trojan detection, the main assumption is the Trojan images have low class entropy when the attacks are successful. But this need not be the case, instead of driving the probability of the target class to 1 and zero entropy, we only need to make sure it is higher than the probability of all the other classes to have successful attacks. Is this a wrong assumption to use?

- In Figure 6 the illustrated Trojan attacks are very noticeable to human eyes. Are they practical?

---

> ### Author Response · Authors · 2024-05-14
> **Response to Reviewer kYEf**
>
> We thank the reviewer for the valuable comments. Please see our responses below:
>
> 1. On the poor writing quality.
>
> We will thoroughly revise the sections, particularly the main contributions, to enhance clarity and readability.
>
> 2. The rationale behind Eqn. 3 is not clear?
>
> We propose mask-based optimization to restrict the search space for optimization, especially for larger images like YouTube Face Recognition datasets. We assume a defense against Badnet-style triggers but can also optimize without masking by imposing sparsity constraints. Figure 7(c) shows the results when the mask is not used for optimization.
>
> 3. What are the definitions of pure accuracy and Trojan accuracy in Table 1? The accuracy under 4x4 and 8x8 triggers are very high. Does that mean that the Trojan attacks are not effective?
>
> Pure accuracy refers to the Trojan model's accuracy with normal test data (without triggers). Ideally, the Trojan model should act
> similarly to the pure model in terms of pure accuracy. Trojan accuracy refers to the Trojan model's accuracy with triggered data. Higher Trojan accuracy indicates a more successful attack.
>
> 4. Table 2 shows the proposed method MTTA performs essentially the same as previous methods like Badnet.
>
> We show in Table 2 that if we present the Trojan data (designed for the MTTA and Badnet models) to a pure model (a model that is not trained with Trojan trigger), it acts normally as the performance of any pure model should not be hampered by the presence of a trigger in the data.  As expected, In Table 2, we show that both MTTA and Badnet triggers do not hamper the accuracy of the pure model. We wanted to highlight that in contrast to MTTA and Badnet style triggering techniques, IA (input-aware Trojan insertion) can make the model vulnerable to early detection (sec 3.4, page 8). Hence, we posit that MTTA and Badnet make better attacks than IA. Between Badnet and MTTA, MTTA is easier to carry out as it does not require a fixed location for putting the trigger. We can think of MTTA as a more resilient version of Badnet as demonstrated in Fig 3(a,c).
>
> 5. In 2.2.2 for Trojan detection, the main assumption is the Trojan images have low-class entropy when the attacks are successful. But this need not be the case, instead of driving the probability of the target class to 1 and zero entropy, we only need to make sure it is higher than the probability of all the other classes to have successful attacks. Is this a wrong assumption to use?
>
> Our entropy measure is computed on the class distribution of the detection dataset instances when added with a reverse-engineered trigger. We just need the predicted class label for each instance, not its prediction probability vector.
>
> 6. In Figure 6 the illustrated Trojan attacks are very noticeable to human eyes. Are they practical?
>
> The trigger size used in Fig 6 is 4x4, which is only 1.6% of the whole image (32x32), hence it should not be that noticeable. We assume that the reviewer is talking about the reverse-engineered triggers of Fig 6 (c,f) where the size of the mask is 16x16, and what we see is that the pattern of the actual 4x4 trigger filled out the whole 16x16 space. This is quite expected because of CNN’s way of pooling features. In short, the reverse-engineered trigger tends to fill up the region of optimization even though the used trigger is small.

---

### Review · Reviewer_agHK · 2024-06-03

**Summary Of Contributions:**

This paper proposes a new version of Badnet, where instead of a single trigger, there are multiple triggers whose locations are not fixed. The authors also propose a defense mechanism against these attacks, demonstrating that their method works effectively.

**Audience:**

No

**Claims And Evidence:**

Yes

**Requested Changes:**

See the weaknesses: Please improve the writing, discuss more advanced method and why your method better

**Strengths And Weaknesses:**

Weaknesses:
Clarity and Readability:
The paper is poorly written and requires significant revision. The current version is not easy to follow, with many sections lacking clarity and coherence. Improving the writing would greatly enhance the paper's readability and accessibility to the audience.
Novelty:

Badnet is an old method, and extending it with multiple, non-fixed location triggers does not introduce significant novelty. The field has evolved with more advanced techniques, and merely modifying Badnet may not contribute substantially to the current state of knowledge.

Currently, there are probabilistic triggers that appear by chance in every image, offering a dynamic approach to trigger placement. The proposed method does not clearly demonstrate superiority over these existing probabilistic trigger methods.
Comparison with Advanced Methods:

There are contemporary methods where the trigger is entirely hidden, and its position is not fixed. These methods offer a more sophisticated approach to backdoor attacks and defenses. The paper does not adequately discuss or compare the advantages of its method over these more advanced techniques. It is crucial to highlight what unique benefits or improvements this method brings to the table.

Experimental Comparison:
While the experiments conducted are solid, they lack comparisons with more advanced topics and state-of-the-art methods. Including a broader range of comparative analyses with recent techniques would strengthen the validity and relevance of the proposed approach.
Strengths:

Addressing Important Topics:
The paper addresses a significant and timely issue in the field of cybersecurity, particularly in the context of machine learning and AI. The idea of enhancing Badnet with multiple triggers and non-fixed locations tackles a relevant problem, aiming to improve the robustness and resilience of models against backdoor attacks.

Experimental Validation:
The experimental results presented in the paper show that the proposed idea is valid. The defense mechanism proposed by the authors demonstrates effectiveness in mitigating the attacks introduced by their modified Badnet. This empirical evidence supports the feasibility and potential impact of their approach.

---

> ### Author Response · Authors · 2024-06-11
> **Response to Reviewer agHK**
>
> We thank the reviewer for the valuable comments. Please see our responses below:
>
> 1.	Clarity and readability
>
> We understand the concern raised by the reviewer. We have identified the root reasons that caused these concerns, and we will thoroughly revise the paper to enhance clarity and readability.
>
> 2.	No significant novelty as Badnet is an old method.
>
> The contributions of the paper are two-fold, one is a more practical backdoor attack and the second is a defense strategy to detect the presence of the proposed attack. We have also shown how advanced the proposed attack method is compared to other stealthy attacks such as Badnet and Input-Aware (dynamic trigger) attacks in Figures 3a-d in the main paper. We are also using a universally applicable entropy-based potential trigger detection strategy other than the existing anomaly detection methods.
>
> 3.	Comparison with Advanced Methods,
>
> We have conducted comparisons with various state-of-the-art methods of backdoor detection [1] and mitigations [2,3] as suggested by the other reviewers to show the potential of the proposed attack and detection method (**Table 1-4**). We have also done more analysis with the **Tiny-Imagenet** (response to reviewer NxLu no: 2) dataset and **vgg11** (Tables 5 and 6) architecture to show the potential of our method. Please see the response to Reviewer 4Cqb and Reviewer NxLu to check the performance comparison.
>
> 4.	Hidden trigger
>
> The proposed attack and the defense strategy fall in the category of visible trigger backdoor attacks and defense. We analysed the proposed settings under this visible trigger category and compared them with the potential state-of-the-art. We have also conducted a new batch of experiments with new architectures and datasets.
>
> [1] FreeEagle: Detecting Complex Neural Trojans in Data-Free Cases, Fu et al., USENIX Security 2023.
>
> [2] Adversarial Neuron Pruning Purifies Backdoored Deep Models, Wu et. al, NeurIPS 2021.
>
> [3] Reconstructive Neuron Pruning for Backdoor Defense, Li et al., ICML 2023.

---

### Decision · Action_Editor_7BLt · 2024-09-08

**Recommendation:** Reject

**Comment:**

The reviewers pointed out this paper has several severe limitations. Specifically, the discussed existing methods in this paper are outdated, which indicates that the proposed method may not be helpful to the current research community.

The issues regarding the experimental settings have not been well addressed in the response. The experiments conducted on 64x64 images are insufficient to demonstrate the proposed method's effectiveness in practical applications. Furthermore, even in such a tiny setting, the Trojan attacks are very noticeable, indicating that the proposed method has altered the image content too much. This suggests that the proposed method is not an effective attack.

In conclusion, the AC believes that the paper does not meet the standards of TMLR. But the AC recommends the authors resubmit this paper after further revisions.

**Audience:**

The individuals who study on the attack and defense of neural network will be interested. However, the paper is not ready for publication in its current form.

**Claims And Evidence:**

The claims in this work should have been supported by experimental results. However, in the current form of this paper, as pointed out by the reviewers, it lacks comparisons with the state-of-the-art method and comparative analyses.  In addition, the presented samples indicate that the proposed method somewhat can not perform decently as expected.

**Resubmission Of Major Revision:**

The authors may consider submitting a major revision at a later time.